# Variations in Biochemical Values under Stress in Children with SARS-CoV-2 Infection

**DOI:** 10.3390/diagnostics12051213

**Published:** 2022-05-12

**Authors:** Alina Belu, Laura Mihaela Trandafir, Elena Țarcă, Elena Cojocaru, Otilia Frăsinariu, Magdalena Stârcea, Mihaela Moscalu, Razvan Calin Tiutiuca, Alina Costina Luca, Anca Galaction

**Affiliations:** 1Sf. Maria Children’s Emergency Hospital, 700309 Iasi, Romania; alina.belu@yahoo.com; 2IOSUD Gheorghe Asachi Technical University, 700050 Iasi, Romania; 3IOSUD “Grigore T. Popa” University of Medicine and Pharmacy, 700115 Iasi, Romania; 4Department of Mother and Child, “Grigore T. Popa” University of Medicine and Pharmacy, 700115 Iasi, Romania; otiliafrasinariu@gmail.com (O.F.); magdabirm@yahoo.com (M.S.); acluca@yahoo.com (A.C.L.); 5Department of Surgery II—Pediatric Surgery, “Grigore T. Popa” University of Medicine and Pharmacy, 700115 Iasi, Romania; 6Department of Morphofunctional Sciences I, “Grigore T. Popa” University of Medicine and Pharmacy, 700115 Iaşi, Romania; elena2.cojocaru@umfiasi.ro; 7Department of Preventive Medicine and Interdisciplinarity, “Grigore T. Popa” University of Medicine and Pharmacy, 700115 Iasi, Romania; mihaela.moscalu@umfiasi.ro; 8Surgical Department, Iacob Czihac Military Emergency Clinical Hospital, 700483 Iasi, Romania; 9Department of Biomedical Sciences, “Grigore T. Popa” University of Medicine and Pharmacy, 700115 Iasi, Romania; anca.galaction@umfiasi.ro

**Keywords:** SARS-CoV-2 infection, children, biochemical markers, risk factors

## Abstract

In the case of SARS-CoV-2 infection, children seem to be less affected than adults, but data regarding epidemiologic characteristics and biochemical values are poor and essentially based on limited case series. The aim of our study is to highlight the predictive value of some biochemical markers at hospitalization, for the correct classification of the patient in the form of disease. Methods: We performed an analytical retrospective study on 82 pediatric patients diagnosed with COVID-19 in the emergency department, with moderate or severe form of disease, and treated in our tertiary hospital. We analyzed the epidemiologic characteristics, symptomatology and biochemical values and compare the data according to the form of disease. Results: The mean age at admission was 4.5 years (median 1 year) and the masculine/feminine ratio was 1.5. Comparing the data between the two groups of patients (42 severe/40 moderate), we observed that the severe form presented with a lower pH at admission (*p* = 0.02), hyperglycemia (*p* = 0.01), increased values of transaminases (*p* = 0.01 and 0.02) and hypoproteinemia (*p* = 0.01). Also, the severe form was statistically significantly associated with comorbidities, acute respiratory distress, rising of the inflammatory markers during hospitalization. Hyperlactatemia (Lactate > 1.5 mmol/L) was significantly associated with the age under one year (*p* < 0.001). Mortality rate was 9.75% and the median age at death was 3 months. Univariate logistic regression model shows that the presence of anemia increased the probability of death 88 times, comorbidities 23.3 times and ketoacidosis 16.4 times. Conclusions: Metabolic acidosis, hyperlactatemia, hyperglycemia, modified hepatic values and hypoproteinemia are biochemical markers associated with the severe form of disease in SARS-CoV-2 infection in children. Presence of anemia, comorbidities and ketoacidosis are important risk factors for death of pediatric patients with SARS-CoV-2 infection.

## 1. Introduction

Since January 2020, the COVID-19 pandemic has spread very fast worldwide. Epidemiological, clinical, biochemical data and management applied in different parts of the globe must be carefully analyzed and the results not yet generalized. And this is due to the extraordinary dynamics of the virus as well as the multifactorial influences related to the specifics of each region and the genetic peculiarities of the affected populations.

The clinical picture of COVID-19 presents different patterns of presentation in adults and children [1,2]. Although children with COVID-19 infection developed milder forms of the disease than those described in adults, they may have severe complications but a low mortality rate [3]. Severe forms of COVID-19 have been found especially in children with congenital heart disease, some genetic syndromes (e.g. sdr Down), malnutrition, obesity, diabetes and cancer. Although the number of cases of COVID-19 infection in children and adolescents has increased recently, many aspects of the clinical features and prognostic factors remain unclear [3,4].

Early diagnosis, identification of cases requiring hospitalization, risk stratification with timely admission to intensive care units and application of appropriate treatment regimens according to current protocols are key elements for the favorable development of COVID-19 in children [5,6]. Current data on clinical forms and prognostic factors in children and adolescents are relatively limited [4]. COVID-19 is not only a disease with respiratory or digestive manifestations, but a multisystemic disease secondary to the complex interaction between immunological, inflammatory and coagulation cascades [7]. In clinical practice early identification of severe forms of COVID-19 in children is essential for the evolution and prognosis of the disease.

In order to predict the evolution of the disease forms from the moment of establishing the diagnosis, the evaluation of some clinical parameters and of some biochemical and immunological markers is absolutely necessary. Which biomarker should be evaluated, when should it be evaluated and what is its usefulness for patient care, are questions that still need to be answered clearly, especially in children? The biochemical markers used in the initial assessment of COVID-19 infections quantify the involvement of various organs or systems, such as: hematological (complete blood count, white blood count (WBC), hemoglobin (Hb), hematocrit (Ht)), inflammation (C reactive protein (CRP), procalcitonin, lactate dehydrogenase ferritin), coagulation (D-dimer levels, fibrinogen, prothrombin time, activated partial thromboplastin time), hepatic values (aspartate aminotransferase (AST), alanine aminotransferase (ALT), bilirubin, albumin), markers of muscle damage (creatine-kinase, myoglobin), renal (serum creatinine), electrolytes, pH [5,6,7,8,9].

The aim of our study is to highlight the predictive value of some biochemical markers at hospitalization, for the correct classification of the patient in the form of disease (moderate or severe). Biochemical markers are valuable for patient management because they can assess the severity or the progression of the disease and can act as risk factors for death.

## 2. Materials and Methods

We conducted a retrospective analytical study of pediatric patients with SARS-CoV-2 infection treated at our tertiary hospital. Prior to data extraction, approval was granted by the Ethics Committee of “Saint Mary” Emergency Children’s Hospital. The analysis included patients who were tested with the real-time Reverse Transcriptase-Polymerase Chain Reaction (RT-PCR) test from nasal and pharyngeal swab samples. The determination of a COVID-19 diagnosis was based on the guidelines by the World Health Organization (WHO) [10]. Inclusion criteria for the study consisted of patients who were hospitalized and treated for SARS-CoV-2 infection from March 2020 to February 2022 in our tertiary care center. The exclusion criteria: patients who resulted as negative for the RT-PCR test, asimptomatic patients with SARS-CoV-2 infection or patients with a mild form of the disease. Patient demographic data, symptoms, comorbidities, biochemical markers, length of hospital stay and mortality were extracted from the hospital’s computer database, analyzed and statistically processed.

We will further define the terms used in the statistical analysis. To compare the data, we divided patients into two age groups (under one year = infants, and over one year of age = child), and also according to the severity of the disease. The severity of the disease was assessed according to the definition and criteria stated by the National Institute of Health, which we reproduce below literally [11,12]:

Moderate illness: Individuals who show evidence of lower respiratory disease during clinical assessment or imaging and who have oxygen saturation (SpO_2_) ≥ 94% on room air at sea level. Severe illness: Individuals who have SpO2 < 94% on room air at sea level, a ratio of arterial partial pressure of oxygen to fraction of inspired oxygen (PaO_2_/FiO_2_) < 300 mm Hg, a respiratory rate >30 breaths/min, or lung infiltrates >50% [11,12].

### Statistical Analysis

We initially performed a descriptive statistical process, depending on the form of the disease, the age group and separately for the deceased. We then tried to find whether there was any link or reciprocal influence between the statistical variables mentioned above (demographics, symptoms, comorbidities, biochemical markers, length of hospital stay and mortality) and the severity of the disease or the age of patients with SARS-CoV-2 infection. First of all, the chi-square test and the contingency tables (cross-tabulation) were used in the analysis; the studied variables were of nominal type. Because the analyzed variables are not normally distributed, in the comparisons made between them depending on the form of the disease (severe/moderate) we used the nonparametric Wilcoxon rank-sum and Mann-Whitney tests. Univariate logistic regression was used to determine the risk of death of patients with SARS-CoV-2 infection; the risk was estimated based on independent variables previously validated by the Pearson chi-square test (χ^2^). In the proposed model, the nominal factor variables were: anemia, comorbidities and ketoacidosis. The dependent variable death is of nominal dichotomous type and can take YES/NO values. All *p*-values less than 0.05 were considered statistically significant and were double-sided. Data are expressed as median, median absolute deviation (MAD), and minimum and maximum values. All calculations were made using standard statistical package (JASP Team (2022). JASP (Version 0.16.1), University of Amsterdam, Amsterdam, The Netherlands, https://jasp-stats.org/, accessed on 1 April 2022).

## 3. Results

From a total of 15,879 persons tested in the analyzed period, 1973 were parents and 104 of them were positive for COVID 19 infection. There were 13,906 children tested and 636 were positive for COVID 19 infection; only 82 of them were eligible for our study (diagnosed with moderate or severe form of SARS-CoV-2 infection). This means that 87% of children tested positive for COVID19 were asymptomatic or with mild forms of disease and 13% were with moderate or severe forms. According to the severity of disease, patients were hospitalized in the pediatric ward to which they belonged or in the intensive care unit if the severity of the condition required it. The mean age at admission was 4.5 years (median 1 year); there were 34 patients under the age of 1 (5 neonates) and 48 patients above 1 year old. The male/female ratio was 1.5.

Comparing the age, length of hospitalization and biochemical values at admission between the two groups of patients (42 with severe form of the disease and 40 with moderate form) in Table 1 and Table 2, we observed that the severe form presented with a lower pH at admission (U = 619, *p* = 0.04), hyperglycemia (U = 517, *p* = 0.02), increased values of transaminases (U = 615, *p* = 0.036 for ALT and U = 622, *p* = 0.043 for AST) and hypoproteinemia (U = 204, *p* = 0.02).

Also, the severe form was statistically significant associated with comorbidities (χ^2^ = 17.852, *p* < 0.001) (Figure 1), acute respiratory distress (χ^2^ = 9.355, *p* = 0.002) and longer hospitalization period (U = 357, *p* < 0.001). We did not find a significant correlation between the severity of the disease and the age group.

When comparing the two groups of age (infants and child), the results obtained were: fever (χ^2^ = 5.896, *p* = 0.02), pneumonia (χ^2^ = 6.185, *p* = 0.02) and comorbidities (χ^2^ = 4.571, *p* = 0.04) were significantly associated with infants. Regarding the biochemical values, hyperlactatemia (Lactate > 1.5 mmol/L) was significantly associated with the under one year of age group (χ^2^ = 14.154, *p* < 0.001); the odds ratio of hyperlactatemia is 8.152 times higher in the age group under one year compared to the child group (Figure 2); high levels of CPR were associated with the child group (χ^2^ = 4.952, *p* = 0.04).

The rate of mortality was 9.75% per total (14.7% in the infant group and 6.25% in the child group; not statistically significant), and the median age at death was 3 months. There were 6 boys and 2 girls. The descriptive statistics for the deceased patients is shown in Table 3. We can see that 5 out of 8 children presented with hyperlactataemia and metabolic acidosis. Hyperglicemia, anemia, hypoproteinemia and increased hepatic transaminases were also observed in the majorities of patients.

Univariate logistic regression was performed to ascertain the effects of anemia, comorbidities and ketoacidosis on the likelihood that patients dying. The univariate logistic regression model was statistically significant, χ^2^ = 28.08, *p* < 0.001 (Table 4).

The model correctly classified 93.9% of cases. The presence of anemia increased the probability of death 88 times, comorbidities 23.3 times and ketoacidosis 16.4 times. In order to estimate the coefficient of determination as a model summary, we used Nagelkerke’s R-square which shows that all three independent variables contributed 61.4% to the risk of death.

Performance Diagnostics: The confusion matrix (Table 5) shows that the 72 true negative and 5 positive cases were predicted by the model. Errors were found in 5 cases (3 cases false negatives and 2 cases false positives).

## 4. Discussion

Despite all the protection measures and vaccination programs, the ongoing COVID-19 pandemic poses various challenges for clinicians [6]. From a clinical point of view, SARS-CoV-2 infection can be asymptomatic or may mimic a mild influenza-like illness, but also can evolve to life-threatening complications, even in children. SARS-CoV-2 affects mainly the respiratory tract resulting in pneumonia, but can also affect gastrointestinal, neurological, or cardiovascular systems [13].

In children, the clinical course of COVID-19 is milder than in adults due to trained immunity, vaccination, frequent respiratory infections, better lung regeneration, and lack of comorbidities [6]. Comorbidities, obesity, smoking, alcohol consumption are often associated with adults with an unfavorable course of disease [14]. In our study we found that 87% of children tested positive for COVID19 were asymptomatic or with mild forms of disease and 13% were with moderate or severe forms, which is different from adults, in which 20% of cases are moderate or severe, as reported by WHO [15,16].

A study conducted in China on 2000 pediatric cases found that only 13% of children infected with COVID-19 were symptomatic; a limitation of the study was that ‘infected’ status was not based on laboratory testing, but only on clinical diagnosis [17]. In a systematic review from 2020, 1124 RT-PCR-confirmed cases from 38 studies were analyzed. According to the severity form of the disease, 14.2% patients were asymptomatic, 36.3% with a mild form, 46.0% moderate, 2.1% severe, and 1.2% patients were critical [18]. It should be noted, however, that these patients have sought medical advice and that the severity of the disease and symptoms may be slightly overestimated [19]. Also in consistent with the literature [16], we found a male preponderance of 1.5.

While reviews have been published on the management and prognostic factors of SARS-CoV-2 infection in adults [13], recent advancements in novel diagnostic and therapeutic methods justify the need for a more comprehensive synthesis of the current literature, in order to elaborate therapeutic protocols and define risk factors in both adults and children. Patient consultation and clinical evaluation are indispensable and mandatory, but for the accuracy of determining the severity of the disease and the subsequent management of the case, certain biological investigations must be performed.

Which biomarker needs to be evaluated, when and in whom, and how best this information can contribute to patient care are questions which currently lack convincing answers [8]. When the anaerobic metabolism of tissues is increased, in infections where there is anemia, fever with increased demand of oxygen, and microvascular obstruction, the value of lactic acid increases as a marker of tissue hypoxia [20]. Changes in biological markers in SARS-CoV-2 infection taken separately are not specific to this disease, but their association may show some specificity. In adults, biochemical markers associated with a severe form of disease are: lymphopenia, neutropenia, elevated serum ALT, AST, LDH, CRP, and ferritin [12,21]. Patients which are critically ill presents with high plasma levels of inflammatory markers, elevated levels of d-dimer and lymphopenia; it is demonstrated that these markers are associated with increased risk of death [13]. We found that metabolic acidosis, hyperlactatemia, hyperglycemia, modified/elevated hepatic values and hypoproteinemia are biochemical markers associated with the severe form of disease in SARS-CoV-2 infection in children; in our study in children, ferritin was not statistically significantly elevated in patients with severe disease. We also found that anemia and ketoacidosis are highly associated with death (increasing the risk of death 88 times and 16.4 times respectively). Therefore, among the biochemical parameters, serum lactate and the presence of metabolic acidosis are useful parameters for appreciating the severity form of illness in children. The value of serum lactate, a parameter of physiological stress and anaerobic metabolism may be recommended in future guidelines to appreciate the COVID-19 severity and the progression of disease. But until now, the value of serum lactate in predicting a severe course in COVID-19 in adults and children is still unclear [22,23].

The first cases of SARS-CoV-2 infection was reported in December 2019; since then there have been over 494 million confirmed cases and 6.1 million deaths reported all around the globe [24]. Elderly patients or those with associated conditions such as cardiovascular disease, obesity, diabetes, chronic respiratory disease, have a more severe course of the disease and a higher risk of death [25]. For children, we demonstrated that the presence of anemia, comorbidities and ketoacidosis are important risk factors for unfavorable evolution of pediatric patients with SARS-CoV-2 infection, and these three independent variables contribute 61.4% to the risk of death. Regarding pediatric patients, general risk factors are represented by certain congenital anomalies such as heart malformations, broncho-pulmonary malformations, malnutrition, low immunodeficiency and age under 3 months [26,27,28]. In our study, the presence of comorbidities increased the risk of death 23.3 times. When comparing the age of our patients, we did not find a significant correlation between the severity of the disease and the age group, but fever, pneumonia and comorbidities were significantly more frequent in the infants group. The rate of mortality was 14.7% for the infant group and 6.25% in the child group, and the median age at death was 3 months; there were 6 boys and 2 girls. Our results are consistent with a multicenter study on 82 institutions across 25 European countries; the study used univariable analysis and found that the most significant factors for requiring intensive care were: age under 1 month, male sex, comorbidities, fever, lower respiratory tract infection, pneumonia or acute respiratory distress syndrome (imagistically diagnosed) and viral co-infection [29]. Although age and concurrent comorbidities of COVID-19 patients largely determine the COVID-19 clinical course, elevated lactate levels may equally facilitate the assessment of disease course. Caterino M and colab. show that is a pathogenic connection between lactic acid and the immune response: high lactate levels being strongly associated with the poor outcome of COVID-19 disease [30].

In terms of recovery, the median duration of hospital stay in adults is 10 to 14 days [31], comparing with 5 days for moderate and 10 days for severe forms in our pediatric cohort study. This difference may again be due to better immunity in children, absence of use of toxic drugs, lower rates of associated abnormalities, and faster recovery.

### 4.1. Limitations of the Study

There are limitations intrinsic to this study because the review was retrospectively performed and it is a single-center study. The small number of cases in total and especially the small group of patients with unfavorable evolution makes it impossible to extrapolate the results obtained in this study to the entire pediatric population.

### 4.2. Strengths of the Study

This study is performed on the largest population lot in our country, as our hospital serves the entire northeastern region of Romania. In mid-2020, this region accounted for 18% of Romania’s population, numbering approximately 4 million people. In any case, we cannot generalize the results obtained in this region to our entire country.

## 5. Conclusions

Metabolic acidosis, hyperlactatemia, hyperglycemia, modified hepatic values and hypoproteinemia are biochemical markers associated with the severe form of disease in SARS-CoV-2 infection in children. When comparing the age of our patients, fever, pneumonia and comorbidities were significantly associated with the infants group. Presence of anemia, comorbidities and ketoacidosis are important risk factors for unfavorable evolution of pediatric patients with SARS-CoV-2 infection. Variations in biochemical values for defining COVID-19 severity or prognosis in children remain an active area of research that may lead to new diagnostic approaches and help us understand disease progression and host responses.

## Figures and Tables

**Figure 1 diagnostics-12-01213-f001:**
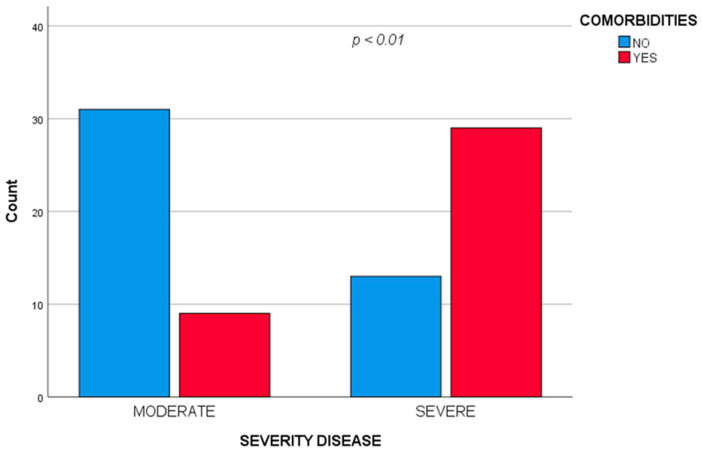
Severe form of disease statistically significant associated with comorbidities.

**Figure 2 diagnostics-12-01213-f002:**
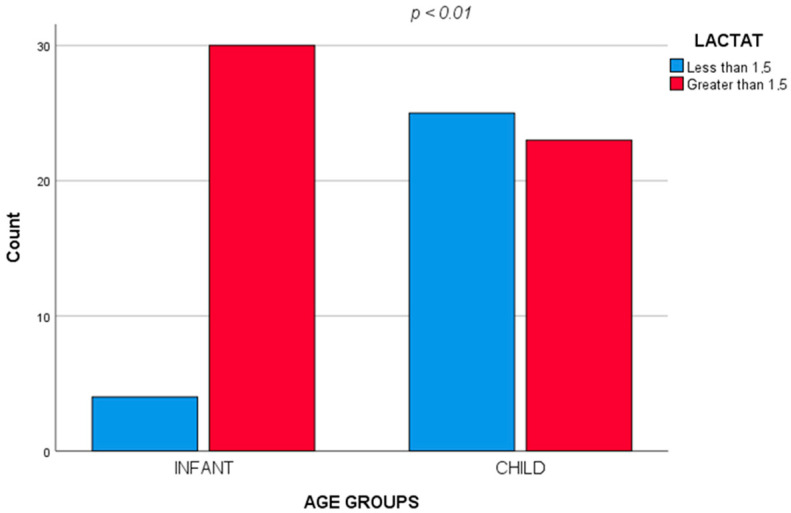
Hyperlactatemia significantly associated with the under one year of age group.

**Table 1 diagnostics-12-01213-t001:** Descriptive Statistics depending on the severity of the disease.

		Valid	Missing	Median	MAD	Minimum	Maximum
Age	MODERATE	40	0	1.000	0.920	0.000	17.000
Age	SEVERE	42	0	2.000	2.000	0.000	17.000
Lenght of Hospit	MODERATE	40	0	5.000	2.000	1.000	16.000
Lenght of Hospit	SEVERE	42	0	10.000	2.500	2.000	36.000
Lactatemia	MODERATE	40	0	2.100	0.800	0.700	4.700
Lactatemia	SEVERE	42	0	2.000	0.800	0.500	14.000
pH	MODERATE	40	0	7.400	0.045	7.100	7.570
pH	SEVERE	42	0	7.365	0.100	7.030	7.570
PaO_2_	MODERATE	40	0	48.000	11.500	21.000	176.000
PaO _2_	SEVERE	42	0	43.000	11.000	17.000	192.000
PaCO_2_	MODERATE	40	0	34.500	8.500	20.000	53.000
PaCO_2_	SEVERE	42	0	39.500	9.500	21.000	97.000
Glycemia	MODERATE	40	0	91.000	9.000	53.000	272.000
Glycemia	SEVERE	42	0	130.500	49.000	53.000	599.000
Hb	MODERATE	40	0	12.100	1.050	8.600	17.100
Hb	SEVERE	42	0	11.350	1.800	6.300	19.400
Ht	MODERATE	40	0	35.650	3.050	24.000	47.710
Ht	SEVERE	42	0	34.900	4.200	17.300	46.000
WBC	MODERATE	40	0	10,385.000	4045.000	1910.000	28,960.000
WBC	SEVERE	42	0	8910.000	5140.000	2030.000	30,980.000
Platelet	MODERATE	40	0	341,500.000	104,000.000	35,800.000	934,000.000
Platelet	SEVERE	42	0	280,500.000	94,000.000	13,000.000	786,000.000
CRP	MODERATE	39	1	2.760	2.110	0.040	118.660
CRP	SEVERE	42	0	4.115	3.425	0.000	173.070
AST	MODERATE	40	0	20.500	6.500	6.000	176.000
AST	SEVERE	42	0	26.500	12.500	8.000	2718.000
ALT	MODERATE	40	0	29.000	8.500	13.000	113.000
ALT	SEVERE	42	0	37.500	18.500	10.000	2724.000
Proteins	MODERATE	18	22	60.785	2.915	45.060	73.580
Proteins	SEVERE	37	5	55.250	7.650	31.920	75.620
Ddimers	MODERATE	17	23	762.000	577.000	34.000	4390.000
Ddimers	SEVERE	30	12	666.500	474.000	78.000	22,176.000
Ferritin	MODERATE	28	12	88.250	46.740	23.300	10,735.250
Ferritin	SEVERE	36	6	132.800	102.720	14.370	3026.650

**Table 2 diagnostics-12-01213-t002:** Comparation between the Age, Lenght of hospitalization and biochemical values at admission depending on the severity form of disease Test Statistics.

	Age	Hospital Stay	Lactatemia	pH	PaO_2_	PaCO_2_	Glycemia	Hb	Ht	WBC	Platelet Count	CRP	ALT	AST	Proteins	Ddimers Ferritin
Mann-Whitney U	722.000	357.000	824.500	619.000	791.500	694.000	517.000	804.500	808.500	814.000	721.500	741.500	615.000	622.000	204.000	219.000 437.000
Wilcoxon W	1542.000	1177.000	1727.500	1522.000	1694.500	1514.000	1337.000	1707.500	1711.500	1717.000	1624.500	1521.500	1435.000	1442.000	907.000	372.000 843.000
Z	−1.098	−4.492	−0.144	−2.052	−0.450	−1.355	−2.997	−0.329	−0.292	−0.241	−1.099	−0.733	−2.088	−2.023	−2.314	−0.797 −0.97
Asymp. Sig. (2-tailed)	0.272	0.000	0.886	0.040	0.653	0.175	0.003	0.742	0.770	0.809	0.272	0.464	0.037	0.043	0.021	0.425 0.365
Exact Sig. (2-tailed)	0.275	0.000	0.888	0.040	0.656	0.177	0.002	0.745	0.773	0.812	0.274	0.467	0.036	0.043	0.020	0.436 0.371

Grouping Variable: SEVERITY FORM DISEASE.

**Table 3 diagnostics-12-01213-t003:** Descriptive statistics for the deceased patients.

Sex	Age (Years)	Days of Hospital.	Lactatemia	pH at Admisssion	PaO_2_	PaCO_2_	Glycemia	Hb	Ht	WBC	Platelet	CRP	ALT	AST	Proteins
M	7.0	7	1.2	7.54	41	44	98	10.3	36.2	30,980	382,000	25.43	45	57	54.88
M	0.3	7	14	7.22	143	27	53	9.7	30.9	12,749	171,000	8.58	255	193	35.68
F	12.0	13	1	7.48	169	30	138	10.9	34.4	15,650	435,000	7.95	105	59	49.5
M	0.1	10	4.4	7.18	53	42	75	8.8	25.4	28,180	370,000	21.12	111	95	68.23
M	0.0	9	4.3	7.24	39	57	251	14.9	46	29,270	13,000	0.68	125	290	31.92
M	0.2	2	2.2	7.34	31	53	222	8.4	25.2	2030	94,000	0.41	130	217	45.33
M	0.3	10	2.2	7.33	22	46	599	8.1	25	30,980	548,000	2.39	65	51	45.03
F	5.0	18	1	7.37	75	38	135	12.3	36.1	6020	276,000	9.37	26	32	65.15

**Table 4 diagnostics-12-01213-t004:** Univariate logistic regression—Model Summary—Death.

Coefficients							
					Wald Test		
	Estimate	Standard Error	Odds Ratio	z	Wald Statistic	df	*p*
(Intercept)	−7.436	2.091	5.898 × 10^−4^	−3.556	12.645	1	<0.001
Anemia (YES)	4.478	1.496	88.056	2.993	8.956	1	0.003
Comorbidities (YES)	3.149	1.460	23.306	2.157	4.653	1	0.031
Ketoacidosis (YES)	2.797	1.357	16.396	2.061	4.247	1	0.039

Death level ‘YES’ coded as class 1.

**Table 5 diagnostics-12-01213-t005:** Confusion matrix.

Observed	Predicted		
	NO	YES	% Correct
NO	72	2	97.297
YES	3	5	62.500
Overall % Correct			93.902

The cut-off value is set to 0.5.

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
