# Peer review of "Variations in Biochemical Values under Stress in Children with SARS-CoV-2 Infection"

_diagnostics, 2022, doi:10.3390/diagnostics12051213_

Round 1
Reviewer 1 Report
In statistics, it indicates that the descriptive will be done with the mean, and it is done with the median. Table 1 must be with mean, not median.
You must justify with references, the criterion of 94 to separate moderate or severe.
Table 2 must be divided into 2 tables, to be better understood. And take the value for two tailed.
line 147 we demonstrated, should be changed by the results obtained,
Author Response
Dear Reviewer,
Thank you very much for evaluating our manuscript. Your recommendations and comments have helped us improve our manuscript. Here we provide the requested corrections and address the comments. The changes we have made in the manuscript are highlighted in red.
Point 1: In statistics, it indicates that the descriptive will be done with the mean, and it is done with the median. Table 1 must be with mean, not median.
Response: Because the analysed variables are not normally distributed, median is more appropriate for characterising our data. We amended the text accordingly.
Point 2: You must justify with references, the criterion of 94 to separate moderate or severe.
Response: The severity of the disease was assessed according to the definition and criteria stated by the National Institute of Health – reference number 11. We also added reference number 12 for severity of illness (clinical course).
Point 3: Table 2 must be divided into 2 tables, to be better understood. And take the value for two tailed.
Response: We formatted the table and took the value for two tailed (we modified the values in the text also).
Point 4: line 147 we demonstrated, should be changed by the results obtained.
Response: We have modified accordingly.
Thank you again for reviewing our manuscript,
Elena Țarcă, MD, PhD
Reviewer 2 Report
The article describes a small number of children affected by COVID 19 infection and the number of markers used is also very small. It is noteworthy that some markers of severity used in adults, such as ferritin, were not included in the study.
Another issue of interest is whether any of the markers found to be elevated in the study offered by the authors offer any specificity in COVID infection or whether they are nonspecific and are elevated in severe pulmonary infections in children treated for other etiologies.
In my overall view, the paper shows weaknesses and I would not accept it for publication in the journal.
Author Response
Dear Reviewer,
Thank you very much for evaluating our manuscript. Your recommendations and comments have helped us improve our manuscript. Here we provide the requested corrections and address the comments. The changes we have made in the manuscript are highlighted in red.
Point 1: The article describes a small number of children affected by COVID 19 infection and the number of markers used is also very small. It is noteworthy that some markers of severity used in adults, such as ferritin, were not included in the study.
Response 1: We agree that there are limitations intrinsic to this study because it is a single-center study. We mentioned that aspect on the special subchapter 4.1. There were 636 children pozitive for SARS-CoV2 infection and we analysed only 82 of them, with moderate and severe forms of disease. But this study is performed on the largest population lot in our country, as our hospital serves the entire northeastern region of Romania. In the future we will try to increase the number of patients in collaboration with other children's hospitals in the country.
Regarding the biological markers studied, there were several, but we tried to present only the most representative ones, so as not to overload the article. We included the analysis for ferritin in table 1 and 2 - The difference was not statistically significant for the severity of the disease.
Point 2: Another issue of interest is whether any of the markers found to be elevated in the study offered by the authors offer any specificity in COVID infection or whether they are nonspecific and are elevated in severe pulmonary infections in children treated for other etiologies.
Response 2: Changes in biological markers in SARS-CoV2 infection taken separately are not specific to this disease, but their association may show some specificity. In this regard, typical laboratory values for COVID-19 (leucocytes <10.0× 109/L, neutrophils <7.0× 109/L, lymphocytes <1.0× 109/L, CRP only moderately elevated (10–130mg/L), procalcitonin <1.0ng/mL) are present [Flick H, Arns BM, Bolitschek J, et al. Management of patients with SARS-CoV-2 infections and of patients with chronic lung diseases during the COVID-19 pandemic (as of 9 May 2020) : Statement of the Austrian Society of Pneumology (ASP). Wien Klin Wochenschr. 2020;132(13-14):365-386. doi:10.1007/s00508-020-01691-0].
We mentioned this in the text.
Similarly to adults, laboratory tests showed an increase in C-reactive protein (CRP) (moderate), transaminases, lactate dehydrogenase, D-dimer and creatine kinase, as well as leukopenia (primarily lymphopenia) [Xu Y, Li X, Zhu B, Liang H, Fang C, Gong Y, et al. Characteristics of pediatric SARS-CoV-2 infection and potential evidence for persistent fecal viral shedding. Nat Med. 2020;26(4):502–5].
In our study, we found that metabolic acidosis, hyperlactatemia, hyperglycemia, modified hepatic values and hypoproteinemia are biochemical markers associated with the severe form of disease in SARS-CoV2 infection in children. By statistical processing methods (logistic regression) we have demonstrated that presence of anemia, comorbidities and ketoacidosis are important risk factors for unfavorable evolution of pediatric patients with SARS-CoV2 infection.
We believe these results are important and worth publishing.
Thank you again for reviewing our manuscript,
Elena Țarcă, MD, PhD
Reviewer 3 Report
Dear authors, enjoy reading your manuscript. I leave some suggestions for your better understanding.
TITLE
The title does not agree with the research or its results. I suggest modifying it. Perhaps to highlight the risk of anemia as a predictor of mortality in this group of patients.
INTRODUCTION
I recommend one or two sentences that put the disease and its relevance in context at the beginning of this section.
The sentence between lines 55 and 57 requires a reference. I suggest using: Tadj A, Lahbib SSM. Our overall current knowledge of COVID 19: an overview. Microbes, Infect Chemother. 2021;1:e1262.
The sentences on lines 66 to 72 seem irrelevant to me. It could be summed up as "multiple tests have been used with different results".
MATERIAL AND METHODS
Mention the data collection period.
RESULTS
The reasons why some cells are highlighted and some data in tables 1, 2 and 4 are placed in red are not understood.
Figures 1 and 2 have to be improved in their presentation and description, in addition to the fact that they can be joined into one.
DISCUSSION
Very well written.
Author Response
Dear Reviewer,
Thank you very much for evaluating our manuscript. Your recommendations and comments have helped us improve our manuscript. Here we provide the requested corrections and address the comments. The changes we have made in the manuscript are highlighted in red.
Point 1: The title does not agree with the research or its results. I suggest modifying it. Perhaps to highlight the risk of anemia as a predictor of mortality in this group of patients.
Response 1: In our study we found that metabolic acidosis, hyperlactatemia, hyperglycemia, modified hepatic values and hypoproteinemia are biochemical markers associated with the severe form of disease in SARS-CoV2 infection in children. Also, anemia and ketoacidosis are important risk factors for unfavorable evolution (death) of pediatric patients with SARS-CoV2 infection. Since all this biochemical markers (and not only anemia) are modified in the severe form of disease (that is, under stress), we think that the title is appropriate.
Point 2: INTRODUCTION: I recommend one or two sentences that put the disease and its relevance in context at the beginning of this section.
Response 2: We have added two sentences in the introduction.
Point 3: The sentence between lines 55 and 57 requires a reference. I suggest using: Tadj A, Lahbib SSM. Our overall current knowledge of COVID 19: an overview. Microbes, Infect Chemother. 2021;1:e1262.
Response 3: We added the suggested reference.
Point 4: The sentences on lines 66 to 72 seem irrelevant to me. It could be summed up as "multiple tests have been used with different results".
Response 4: These biological markers mentioned here were in fact the ones used in our analysis, so we decided to list them.
Point 5: MATERIAL AND METHODS: Mention the data collection period.
Response 5: The period is mentioned here: „Inclusion criteria for the study consisted of patients who were hospitalized and treated for SARS-CoV2 infection from March 2020 to February 2022 in our tertiary care center.” – We highlighted in the text, also.
Point 6: RESULTS: The reasons why some cells are highlighted and some data in tables 1, 2 and 4 are placed in red are not understood.
Response 6: In table 1 we highlighted the data for the severe form of disease, for a better vizualization. The data placed in red (for statistically significance) were modified (turned black).
Point 7: Figures 1 and 2 have to be improved in their presentation and description, in addition to the fact that they can be joined into one.
Response 7: We replaced the Figure 1 and 2.
Point 8: DISCUSSION: Very well written.
Response 8: Thank you.
Thank you again for reviewing our manuscript,
Elena Țarcă, MD, PhD
Round 2
Reviewer 2 Report
The article submitted for review has improved with the modifications suggested to the authors, which is why I would accept it for publication without modifications.